# Brain Endothelial Cells Activate Neuroinflammatory Pathways in Response to Early Cerebral Small Vessel Disease (CSVD) Patients’ Plasma

**DOI:** 10.3390/biomedicines11113055

**Published:** 2023-11-14

**Authors:** Adriana Cifù, Francesco Janes, Catia Mio, Rossana Domenis, Maria Elena Pessa, Riccardo Garbo, Francesco Curcio, Mariarosaria Valente, Martina Fabris

**Affiliations:** 1Department of Medicine (DAME), University of Udine, 33100 Udine, Italy; adriana.cifu@uniud.it (A.C.); catia.mio@uniud.it (C.M.); rossana.domenis@uniud.it (R.D.); francesco.curcio@asufc.sanita.fvg.it (F.C.); mariarosaria.valente@asufc.sanita.fvg.it (M.V.); martina.fabris@asufc.sanita.fvg.it (M.F.); 2Department of Head, Neck and Neuroscience, Azienda Sanitaria Universitaria Friuli Centrale (ASUFC), 33100 Udine, Italy; mariaelena.pessa@asufc.sanita.fvg.it (M.E.P.); riccardo.garbo@outlook.it (R.G.); 3Neurology Unit of Gorizia-Monfalcone, Azienda Sanitaria Universitaria Giuliano-Isontina (ASUGI), 34100 Gorizia, Italy; 4Institute of Clinical Pathology, Azienda Sanitaria Universitaria Friuli Centrale (ASUFC), 33100 Udine, Italy

**Keywords:** neuroinflammation, cytokines, endothelial dysfunction, cerebral small vessel disease

## Abstract

The pathogenesis of cerebral small vessel disease (CSVD) is largely unknown. Endothelial disfunction has been suggested as the turning point in CSVD development. In this study, we tested the effect of plasma from CSVD patients on human cerebral microvascular endothelial cells with the aim of describing the pattern of endothelial activation. Plasma samples from three groups of young subjects have been tested: PTs (subjects affected by early stage CSVD); CTRLs (control subjects without abnormalities at MRI scanning); BDs (blood donors). Human Brain Endothelial Cells 5i (HBEC5i) were treated with plasma and total RNA was extracted. RNAs were pooled to reduce gene expression-based variability and NGS analysis was performed. Differentially expressed genes were highlighted comparing PTs, CTRLs and BDs with HBEC5i untreated cells. No significantly altered pathway was evaluated in BD-related treatment. Regulation of p38 MAPK cascade (GO:1900744) was the only pathway altered in CTRL-related treatment. Indeed, 36 different biological processes turned out to be deregulated after PT treatment of HBEC5i, i.e., the cytokine-mediated signaling pathway (GO:0019221). Endothelial cells activate inflammatory pathways in response to stimuli from CSVD patients’ plasma, suggesting the pathogenetic role of neuroinflammation from the early asymptomatic phases of cerebrovascular disease.

## 1. Introduction

Cerebral small vessel disease (CSVD) is a nosological entity that gathers some diseases affecting the small arteries, capillaries, arterioles, and venules of the brain. CSVD is one of the most common types of cerebrovascular diseases [1]. With a prevalence that increases exponentially with aging, it is one of the major risk factors for both acute stroke and for cognitive decline [2]. Frequently, the evidence of CSVD is incidentally established after brain imaging. Many patients, in fact, have only mild signs at a neurological examination or just subtle, often subjective, symptoms of neurocognitive dysfunction (e.g., isolated dysexecutive syndromes or memory impairment), recurrent dizziness, or mood changes. It is noteworthy that those symptoms represent relevant prodromal features of the early stages of neurodegenerative disorders (such as Alzheimer’s disease), and of the aging process as well [3].

Although rough features of CSVD are visible at standard brain CT scans, MRI is far more sensitive in detecting above all the total burden of brain lesions, and those with a hemorrhagic nature. The characteristic abnormalities observed in brain MRI include lacunar infarcts (i.e., infarcts of small size, with a diameter < 1 cm, often asymptomatic), white matter hyperintensities (WMH), periventricular hyperintensity (PV), cerebral microbleeds (CMB), enlarged perivascular spaces and brain atrophy [4,5]. 

Cerebral small vessel disease can be caused by several pathological processes and is highly variable in terms of clinical presentation and rate of disease progression. The two main pathological processes involved in the development of CSVD are arteriosclerosis and cerebral amyloid angiopathy, which cause non-amyloid-related and amyloid-related CSVD, respectively. Several studies suggest, nevertheless, that there is a wide overlap between these two forms [4,6]. Some inherited monogenic diseases are associated with CSVD, such as CADASIL (cerebral autosomal dominant arteriopathy with subcortical ischemic strokes and leukoencephalopathy; caused by *NOTCH3* mutations), *COL4A1* and *COL4A2* mutations associated syndromes, Fabry disease (due to mutations in *a-GAL*), RVCL (retinal vasculopathy with cerebral leukodystrophy; caused by *TREX1* mutations), CARASIL (cerebral autosomal recessive arteriopathy with subcortical infarcts and leukoencephalopathy; caused by *HTRT1* mutations), HTRA1-AD disease (due to *HTRA1* mutations), and MELAS (mitochondrial encephalopathy, lactic acidosis, and stroke-like episodes). However, the prevalence of monogenic disorders in CSVD patients seems very low if compared to sporadic CSVD [5]. The role of not completely known acquired causes (metabolic, environmental, inflammatory, etc.) of sporadic CSVD is even more supported by its association with several heterogenous disorders, such as Alzheimer’s disease, Parkinson’s disease, multiple sclerosis, and rheumatologic diseases; and also with chronic conditions, such as chronic hypoxia, obesity, diabetes, congestive heart failure, and metabolic syndrome [4,5,7,8,9]. The pathogenesis of CSVD is largely unknown and consequently tailored preventive treatments are currently unavailable [2,4]. There is evidence that conventional vascular risk factors, above all hypertension, and microatherosclerosis of small arteries play a pivotal role, above all in the elderly population. Consequently, traditional preventive treatment regimens based on combination of antiaggregants, anticoagulants, lipid-lowering and antihypertension drugs are reasonable in those patients. However, sporadic CSVD is still nowadays not considered a definite indication to a primary prevention treatment, unless in the context of a high vascular risk. In fact, non-atherosclerotic acquired or genetically determined structural abnormalities in the extracellular matrix have been demonstrated in several CSVD forms [10]. CSVD patients show abnormal cerebral small vessel structure, associated with a dysfunctional endothelium [11]. A relatively recent hypothesis refers to a primary endothelial dysfunction and subsequent blood–brain barrier dysfunction as the turning point of its early stages. Indeed, studies with animal models of CSVD have confirmed that endothelial dysfunction is the first alteration that leads to the development of this disease [4]. Rajani et al. used a rat model of SVD and showed that dysfunctional Endothelial Cells (ECs) contribute to altered myelination through the release of heat shock protein 90α, which blocks oligodendroglial differentiation [12]. 

Conventional atherosclerosis, systemic inflammation, perivascular space abnormalities, as well as glymphatic system disruption, all seem to converge towards blood–brain barrier dysfunction and finally, progressively, towards failure of vascular homeostasis [13,14,15,16]. Given this complex and partially unknown background, and despite several studies explored this field, it is not unexpected that we still not know which pathway/s is/are chronically and firstly activated. That information would nevertheless be essential to develop new drug treatments and prevention strategies in cerebrovascular diseases [4]. 

In our previous study [17], we evaluated a series of biomarkers in a cohort of young asymptomatic patients with CSVD. We demonstrated that in this group of patients only ADMA (Asymmetric Di-Methyl Arginine) appeared to be a significant marker, suggesting that endothelial dysfunction might play a central role in asymptomatic SVD. Considering these results, in this study, we aimed to investigate the global effects of plasma from asymptomatic young CSVD patients on human cerebral microvascular endothelial cell molecular expression with particular attention to pathways of endothelial activation and inflammation. 

## 2. Materials and Methods

### 2.1. Patients’ Sample Selection Process

This study was performed on a subgroup of plasma samples derived from the previous study [17], selecting 9 patients with CSVD (PT: 8 females and 1 male; mean age 50.6 ± 7.7 years) without a history of cerebrovascular events and with less than 2 vascular risk factors and 7 control subjects from the same study without CSVD or other abnormalities at MRI scanning (CTRLs: 5 females and 2 males; mean age 51.0 ± 14.9 years). We also added 6 blood donors (BDs: 4 females and 2 males; mean age 33.3 ± 18.5 years) followed at our Transfusion Medicine Department.

Whole blood of patients and controls were collected in EDTA-treated tubes. Plasma was obtained by centrifugation for 15 min at 2000× *g* using a refrigerated centrifuge. After centrifugation, plasma samples were stored at −80 °C until use. 

### 2.2. Human Brain Endothelial Cells Culture and Treatments

Human Brain Endothelial Cells 5i (HBEC5i, ATCC^®^ CRL3245™,Manassas, Virginia, USA) were maintained in vessels coated with 0.1% Gelatin (ATCC^®^ No. PCS999027, Manassas, Virginia, USA) at 37 °C in 5% CO_2_ atmosphere in DMEM: F12 (D8437, Merk KGaA, Darmstadt, Germany) supplemented with 40 μg/mL endothelial growth supplement (ECGS; E2759 Merk KGaA, Darmstadt, Germany), Fetal Bovine Serum (FBS, Merk KGaA, Darmstadt, Germany) to a final concentration of 10% and Penicillin–Streptomycin solution (P4333, Merk KGaA, Darmstadt, Germany) to a final concentration of 1%.

HBEC5i cells were seeded at a density of 15,000 cells/cm^2^ and, the day after, the culture medium was replaced with fresh medium supplemented with Penicillin–Streptomycin solution to a final concentration of 1% and 15% *v*/*v* of plasma from CSVD patients (*n* = 9), control (*n* = 7) or blood donors (*n* = 6).

Cells were treated for 24 h. For untreated cells, the culture medium was replaced with fresh medium supplemented with Penicillin–Streptomycin solution to a final concentration of 1% and FBS to a final concentration of 10%.

After treatments, HBEC5i cells were washed twice with Dulbecco′s Phosphate-Buffered Saline w/o phenol red (D-PBS, D8537 Merk KGaA, Darmstadt, Germany).

Cells were detached with trypsin 0.05%-EDTA 0.02% solution (ECB3052D, Euroclone, Milano, Italy) and centrifuged for 5 min at 800× *g*. 

The cell viability was determined using the trypan blue (T6146, Merk KGaA, Darmstadt, Germany) exclusion method [18]. 

### 2.3. RNA Isolation, Library Preparation and Next Generation Sequencing (NGS)

Total RNA was extracted from cells using the RNeasy Kits (QIAGEN, Hilden, Germania), according to the manufacturer’s protocol. Extracted RNA was quantified using the Qubit RNA high-sensitivity assay kit (Q32852, Thermo Fisher Scientific, Waltham, Massachusetts, USA), quality tested by Agilent 2100 Bioanalyzer RNA Nano assay (Agilent technologies, Santa Clara, CA, USA) and stored at −80 °C until use. 

RNA from multiple biological replicates (9 PTs, 5 CTRLs, 4 BDs) were pooled prior sequencing to reduce gene expression-based variability as described in: [19]. 

The Universal Plus mRNA-Seq kit (Tecan Genomics, Redwood City, CA, USA) was used for library preparation, following the manufacturer’s instructions, starting with 200 ng of good-quality RNA (R.I.N. > 7) as input. Final libraries were quantified by using the Qubit 2.0 Fluorometer (Thermo Fisher Scientific) and quality tested by Agilent 2100 Bioanalyzer DNA High Sensitivity assay (Agilent Technologies, Santa Clara, CA, USA). Libraries were then processed with Illumina cBot for cluster generation on the flowcell, following the manufacturer’s instructions and sequenced on the paired-end 150 bp mode on NovaSeq 6000 (Illumina, San Diego, CA, USA). The CASAVA 1.8.2 version of the Illumina pipeline was used to processed raw data for both format conversion and de-multiplexing.

### 2.4. RNA-Seq Bioinformatics Analysis

Raw sequences were analyzed as previously described [20]. Briefly, sequencing read quality was evaluated using the ShortRead (v1.44.3) R/Bioconductor package [21]. Quality, adapter and contamination filtering were performed using the Trimmomatic [22] command-line tool. Processed reads were aligned to the human reference genome GRCh37/hg19 and differentially expressed genes were identified using the DESeq2 (v1.26.0) R/Bioconductor package [23], considering statistically significant results with a log2 fold change (FC) ≥ 1.5 and a FDR-adjusted *p*-value (i.e., q-value) ≤ 0.05. Extended gene annotations (including HGNC gene symbol, description and transcript type) were obtained using the biomaRt (v2.42.0) R/Bioconductor package [24]. Heatmaps were obtained by processing sequencing data with the online tool Morpheus (https://software.broadinstitute.org/morpheus (accessed on 29 June 2023)). Pathway analysis was performed with the web tool GOrilla (http://cbl-gorilla.cs.technion.ac.il/ (accessed on 29 June 2023)), setting a FDR-adjusted *p*-value ≤ 0.05 for significance. 

### 2.5. Target Gene Expression Analysis

To confirm data obtained with sequencing, qPCR was performed. Complementary DNA (cDNA) was generated using the SuperScript™ IV VILO™ Master Mix (11756050, Thermo Fisher Scientific) using 2.5 µg of total RNA, according to the manufacturer’s protocols. mRNA levels of *IL12A* (Interleuchin12A), *CXCL8* (C-X-C Motif Chemokine Ligand 8/IL-8), *LIF* (Leukemia Inhibitory Factor) were assessed by quantitative PCR (qPCR) using SsoAdvance Universal SYBR green super mix (Bio-Rad Laboratories, Hercules, CA, USA) on a QuantStudio 3 System (Applied Biosystems, Waltham, MA, USA). The QuantStudio Design and Analysis software v1.5.0 (Applied Biosystems), was used to calculate mRNA levels with the 2^−∆∆Ct^ method, and *GAPDH* was used as reference. Oligonucleotide primers were purchased from Merk KGaA. The sequences of the primers used in this study are the following: *IL12A* forward 5′ CAA GAC CAT GAA TGC AAA GC3’ and reverse 5’CTG CCA GCA TGT TTT GAT CT 3’; *CXCL8* forward 5′CTG ATT TCT GCA GCT CTG TG3′ and reverse 5′GGG TGG AAA GGT TTG GAG TAT G3′; *LIF* forward 5′GAA CCA GAT CAG GAG CCA ACT G 3′ and reverse 5′CCA CAT AGC TTG TCC AGG TTG TT3′; *GAPDH* forward 5′AGT ATG ACA ACA GCC TCA AG 3′ and reverse 5′ TCT AGA CGG CAG GTC AGG TCC AC 3′.

All experiments were performed in triplicate.

### 2.6. Statistical Analysis

Statistical analysis was performed using GraphPad Prism Software v.5 (GraphPad Software, Boston, MA, USA). Data were tested for normal distribution using the Kolmogorov–Smirnov test. After assessing the absence of a normal distribution, repeated measurements were analyzed by t-Student analysis followed by the Wilcoxon post-test for viability and one-way analysis of variance (ANOVA) followed by the Dunnett post-test for relative gene expression analysis. *p* values less than 0.05 were considered significant. Quantitative variables were expressed as the mean ± standard deviation (SD). 

## 3. Results

### 3.1. Treatments with Plasma

To evaluate the hypothesis of the key role of endothelial dysfunction in CSVD, HBEC5i cells were treated for 24 h with plasma from either CSVD patients (PTs), controls (CTRLs) or blood donors (BDs). We enrolled as controls age- and sex- matched subjects experiencing and undergoing brain MRI that displayed normal scans. Moreover, other neurological, vascular, and chronic inflammatory diseases were excluded during a clinical interview [17]. To set up the experiments and rule out any possible cellular activation mediated by plasma itself, plasma from blood donors was also used as a control. 

To assess whether treatments with human plasma affected cell viability, HBEC5i cells were treated with plasma from blood donors, controls, and patients (final concentration 15% *v*/*v*) for 24 h and viability was assessed by trypan blue assay (*n* = 3). As shown in Figure 1, none of the treatments significantly affected cell viability.

### 3.2. Study of the Effects of CSVD Plasma Incubation on Molecular Expression in HBEC5i Cells

To evaluate the possible effects of plasma from CSVD patients at the molecular mechanism with particular attention to pathways related to endothelial dysfunction, RNA sequencing was performed. HBEC5i cells were treated for 24 h as previously described in the methods and total RNA was extracted. Replicates from the same treatment were pooled together and then underwent sequencing. 

As represented in the heat map in Figure 2, HBEC5i showed a quite different response depending on the treatment performed.

After removing low-quantity reads, 219, 175 and 174 differentially expressed genes were highlighted comparing treatment with plasma of PTs, CTRLs and BDs HBEC5i untreated cells (at log2 fold change ≥ 1.5). A total of 115 transcripts were up-regulated, and 104 transcripts were down-regulated in PTs-treated cells; 84 transcripts were up-regulated, and 91 transcripts were down-regulated in CTRL-treated cells; 87 transcripts were up-regulated, and 87 transcripts were down-regulated in BD-treated cells.

Differentially expressed genes were then subjected to pathway analysis to outline which pathways were mostly affected by each treatment. Pathways commonly regulated were excluded from the analysis. No significantly altered pathway was evaluated in BD-related treatment. Regulation of p38 MAPK cascade (GO:1900744) was the only pathway altered in CTRLs-related treatment. Indeed, 36 different biological processes turned out to be deregulated after PTs plasma treatment of HBEC5i. Data are represented in Table 1.

Then, we focused on the cytokine-mediated signaling pathway (GO:0019221). From this pathway, we selected three representative cytokines involved in neuroinflammation and mRNA levels of *CXCL8*, *LIF* and *IL12A* were assessed by qPCR to confirm RNA-seq-related up-regulation. Indeed, as shown in Figure 3, compared to untreated cells, all three transcripts turned out to be up-regulated in treated cells. 

## 4. Discussion

In this study, we showed that treatment of brain endothelial cells with plasma from patients with “non-lacunar, asymptomatic CSVD” alters several molecular pathways, while treatment with plasma from controls or blood donors has no significant effect on brain endothelial cells. 

Notably, patients’ plasma up-regulates cytokine-mediated signaling pathways likely involved in neuroinflammation. This suggests that in those patients, a low-grade inflammation at the systemic level may be responsible for stressed endothelial cells at the cerebral level.

Although this research does not clarify which molecules specifically act on endothelial cells, it highlights the link between a pro-inflammatory phenotype and the early asymptomatic stages of cerebrovascular disease; moreover, it suggests that one or more specific biomarkers of endothelial activation and inflammation could be detected directly in the blood during the early stages of disease.

In recent decades, the link between inflammation and CSVD has already been investigated and proposed as one of the key mainstays of its pathogenesis [25]. Beyond brain imaging, blood-derived biomarkers have been investigated, both in the cerebrospinal fluid and in the plasma [26,27,28], but they are still poorly characterized. Indexes of overall immune status, such as the “neutrophil-to-lymphocytes ratio” (NFL) and “systemic immune-inflammation” (SII), proved to be positively correlated with some CSVD features, although this evidence is only partially consistent all through the literature [29]. Mid-regional pro-adrenomedulin (MR-proADM) has been proposed as a biomarker of CSVD progression in a longitudinal observational study [30]. Systemic traditional inflammation runs parallel with immune cells trained by atherogenic substance exposure, perpetuating a proinflammatory phenotype: recent research showed that cytokine release is durably enhanced in established atherosclerosis and that cytokines such as CXCL8 and IL-17 are preferential biomarkers of trained immunity [31]. 

The cytokine-mediated signaling pathways and their consequent effects on cells, on diseases occurrence and eventually differential phenotyping, still represent an evolving field, above all in neurological disorders. To the best of our knowledge, there is very little evidence of the role of the specific cytokines and proteins we found overexpressed in brain endothelial cells after treatment with CSVD plasma patients. 

CXCL8 is a widely studied chemokine, with several effects on cell interactions: it is chemoattractive for neutrophil, chemotactic for endothelial cells, and stimulates stem cells, thus proving to be involved in acute inflammation and angiogenesis [32]. It showed several effects within the CNS as well, although often not completely understood, on proliferation, migration, survival of and interaction between neurons and glial cells. Some of these effects have been proposed as distinctive elements in selective inflammatory diseases of the CNS such as multiple sclerosis [33], but also in neuroinflammation associated to neurodegenerative diseases such as Alzheimer’s disease [34]. Few studies also explored CXCL8 signaling in cerebrovascular disorders. Zheng et al. demonstrated, in a microarray analysis, that CXCL8 overexpression (along with TNF, SOC3 and TNFAIP3) could serve as a biomarker of both coronary artery disease (CAD) and ischemic stroke (IS) occurrence [35]. Moreover, in a rat model of acute stroke, inhibition of CXCL8 receptor with reparixin improved neurological outcomes along with a reduction in inflammatory response [36]. Inconsistent results in patients with sickle cell anemia, in which a poor outcome (i.e., number of recurrent VOC) seems to depend on high levels of IL-6, rather than CXCL8 [37], likely underline once more the heterogeneity of cerebrovascular diseases. 

Leukemia Inhibitory Factor (LIF) is an anti-inflammatory cytokine that acts in a signaling cascade that promotes cell survival and neuroprotection through the MAPK, PI3K/Akt and JAK/STAT pathways. The activation of this cascade has been demonstrated both in rat models of stroke and in cultured neurons [38,39]. LIF receptor expression on plasma membrane occurring after a brain injury is also determinant in enhancing neuroprotective effect [39]. LIF overexpression in our treated endothelial cells suggests that those pathways could be activated also in CSVD, possibly trying to counteract endothelial damage. 

A mouse and in vitro study recently explored the mechanism of microglia chemotaxis within ischemic brain tissue. IL12A is suggested to contribute to chemotactic signaling for neutrophils, together with other chemokines, soon after brain ischemia [40].

The vascular endothelial growth factor A (VEGF-A), under normal conditions, is neuroprotective with direct effects on neural cells and their progenitors and indirect effects on cerebral perfusion. A pathological increase in VEGF-A levels is involved in enhanced vessel permeability and leakage and altered blood–brain barrier integrity, as in demyelinating diseases [41]. Overexpression of VEGF-A in our model could be related to the failure of vascular homeostasis and subsequent blood–brain barrier dysfunction. Chronic hypoxia-ischemia is a characteristic feature of both acute brain injury and chronic neurodegenerative diseases and may contribute to cerebral small vessel disease pathogenesis and development of WMH [8,42]. It is important to identify markers of hypoxic suffering. An in vitro study demonstrated an increase in heme oxygenase-1 (HMOX1) mRNA levels, after 8 h of hypoxia [42]. In the future VEGF, HMOX-1 and HIF1α (hypoxia-inducible factor 1α) could be detect in blood of patients affected by CSVD, to assess a hypoxic state.

Heat shock protein 90β (HSP90B1) is a chaperone protein, a member of the heat shock protein 90 family and is a highly expressed in all types of brain cells. In Alzheimer’s disease, HSP proteins play a neuroprotective role by inducing microglial activation by facilitating Aβ clearance and cytokine production. Overexpression of HSP can be induced by stress conditions, such as ischemia and hypoxia [43]. In a rat model of CSVD, secretion of heat shock protein 90α is associated with dysfunctional brain endothelium [12]. 

Granulocyte-macrophage colony-stimulating factor (GM-CSF) is a pro-inflammatory cytokine that leads to induce the differentiation of granulocytes and monocytes. In patients with ischemic stroke, serum GM-CSF levels are significantly higher than those in healthy individuals, and they correlate with the severity of neurological symptoms 12 h after stroke onset, moreover Ota et al. demonstrated that serum GM-CSF levels correlated positively with the severity of WMH in MPA patients [44]. Then, levels of serum GM-CSF could be associated with cerebrovascular lesions.

Revealing CSVD pathogenesis is relevant also for new treatment opportunities. A group of promising new molecules, such as the supplementation of L-Arginine, L-acetyl-cisteine, dietary flavonoids, and a higher proportion of proteins in diet itself, proved to ameliorate endothelial function, by acting on oxidative stress pathways and indirectly on neuroinflammation [45,46,47,48]. Other compounds, instead, such as reparixin [35], dauricin [39] and LIF [37], are under investigation as direct specific modulators of neuroinflammatory response. Studies that explored compounds preserving endothelial function in chronic neurovascular disorders face above all the difficulty of a long follow-up to see a reliable clinical outcome. In this regard, the availability of solid blood-based biomarkers could both identify very early stages of the disease and identify precociously patients at risk of progression. This in turn could help to validate additional drug treatment for CSVD and enable also to track the effect of new drugs within a shorter clinical follow-up. A few limitations of this study must be mentioned: i. the indirect evidence of a neuroinflammatory response in early CVSD as sustained by plasma effect on endothelial cells, but without a direct confirmation of biomarkers’ levels change in plasma; ii. The small sample size, and finally iii. The gender imbalance of the samples we analyzed. We incurred a gender imbalance because we herein chose to privilege the analysis of plasma samples which displayed—in our previous study—higher levels of biomarkers related to endothelial damage. However, previous studies suggest that males and females express different inflammatory patterns under physiological condition [49] and in response to brain ischemic insult [50]. Overall, these issues limit the generalization of our results to both sexes, and do not allow conclusive statements on pathogenetic mechanisms in early CSVD. However, we think that the data we show herein will be helpful in further studying clinically relevant blood biomarkers of CSVD.

## 5. Conclusions

Consistently with the little evidence published in acute stroke animal and in vitro models, our study suggests that endothelial cells activate inflammatory pathways in response to CSVD patients’ plasma. Since previous studies indicate that the up-regulated patterns we found in cerebral endothelial cells are globally non-protective, similarly to acute ischemic stroke, we propose that neuroinflammatory pathways are already activated in the early and asymptomatic phases of cerebrovascular disease.

Biomarkers of neuroinflammation could be detected earlier, longitudinally, and directly in the blood of aging or at-risk populations, thus possibly allowing a more specific pharmacological treatment than traditional vascular preventive drugs. 

Further research is needed to overcome limitations of this and other studies, above all trying to consider the high heterogeneity of CSVD and the lack of longitudinal assessment of inflammatory response. 

## Figures and Tables

**Figure 1 biomedicines-11-03055-f001:**
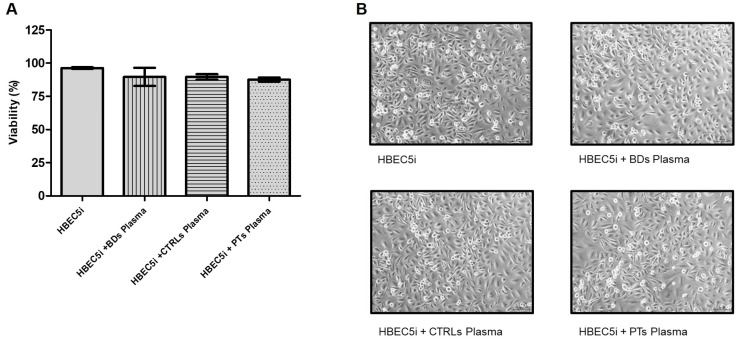
(**A**) Cell viability was determined by trypan blue exclusion assay (*n* = 3). Columns, mean; bars, SD. Treatment with plasma did not induce significant difference compared to untreated cells. (**B**) Representative phase contrast images (10× magnification) of HBEC5i untreated or treated with 15% *v*/*v* of human plasma for 24 h (t-Student analysis and the Wilcoxon post-test).

**Figure 2 biomedicines-11-03055-f002:**
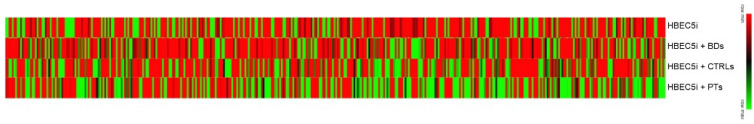
Heatmap showing the differentially expressed genes comparing treatment with PTs, CTRLs and BDs plasma with HBEC5i untreated cells, respectively (at log2 fold change ≥ 1.5; see Methods for further details).

**Figure 3 biomedicines-11-03055-f003:**
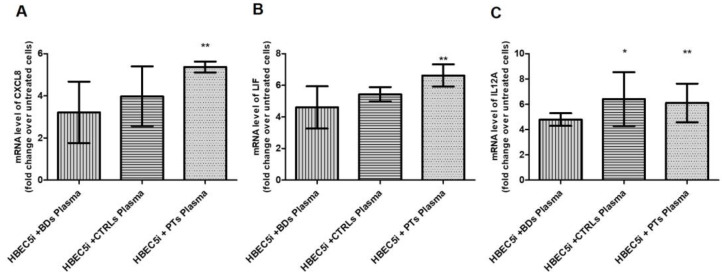
HBEC5i cells were treated with 15% *v*/*v* of human plasma of BDs, CTRLs and PTs for 24 h. The relative expression of CXCL8 (**A**), LIF (**B**) and IL12A (**C**) was analyzed by qPCR (*n* = 4). Data were expressed as the fold change over untreated cells. Columns, mean; bars, SD; * significant difference *p* < 0.05, ** significant difference *p* < 0.01, ANOVA and the Dunnett post-test.

**Table 1 biomedicines-11-03055-t001:** De-regulated pathways by CSVD treatment compared to untreated HBEC5i (only differentially regulated pathways in cells treated with plasma of patients are enlisted, *p* < 0.05).

GO Term	Description
GO:0036500	ATF6-mediated unfolded protein response
GO:0007050	cell cycle arrest
GO:0070887	cellular response to chemical stimulus
GO:0071310	cellular response to organic substance
GO:0051716	cellular response to stimulus
GO:0033554	cellular response to stress
GO:0019221	cytokine-mediated signaling pathway
GO:0001892	embryonic placenta development
GO:0030968	endoplasmic reticulum unfolded protein response
GO:0043066	negative regulation of apoptotic process
GO:0010648	negative regulation of cell communication
GO:0060548	negative regulation of cell death
GO:0043069	negative regulation of programmed cell death
GO:0048585	negative regulation of response to stimulus
GO:0023057	negative regulation of signaling
GO:1901564	organonitrogen compound metabolic process
GO:1990440	positive regulation of transcription from RNA polymerase II promoter in response to endoplasmic reticulum stress
GO:0036003	positive regulation of transcription from RNA polymerase II promoter in response to stress
GO:0042981	regulation of apoptotic process
GO:0010646	regulation of cell communication
GO:0051726	regulation of cell cycle
GO:0031326	regulation of cellular biosynthetic process
GO:0080135	regulation of cellular response to stress
GO:0043067	regulation of programmed cell death
GO:0031399	regulation of protein modification process
GO:1905897	regulation of response to endoplasmic reticulum stress
GO:0080134	regulation of response to stress
GO:0023051	regulation of signaling
GO:0042762	regulation of sulfur metabolic process
GO:1901342	regulation of vasculature development
GO:0042221	response to chemical
GO:0034976	response to endoplasmic reticulum stress
GO:0010033	response to organic substance
GO:0006950	response to stress
GO:0035966	response to topologically incorrect protein
GO:0006986	response to unfolded protein

## Data Availability

Data are contained within the article.

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
