# Peer review of "Brain Endothelial Cells Activate Neuroinflammatory Pathways in Response to Early Cerebral Small Vessel Disease (CSVD) Patients’ Plasma"

_biomedicines, 2023, doi:10.3390/biomedicines11113055_

Round 1
Reviewer 1 Report
Comments and Suggestions for Authors
Author came up with an interesting observation of brain endothelial cell mediated neuroinflammatory pathway activation to early CSVD in patient's plasma.
I have few minor concerns,
1. Sample size is very small and there is huge variability, for instance, females are outnumbered than male (PTs 9-1, CTs, 5-5, BD, %-2). Studies have shown that neuroinflammatory pathways varied based on sex. At this point we cannot make any statements based on small sample size.
2. Figure 2 is very hard to see, please try to improve the quality of image and make sure its legends are readable.
3. Based on RNA-seq, author explored qPCR of CXCL8, LIF and IL-12A, but why the author did not evaluate concentration using ELISA? An increased mRNA expression does not confirm an increased protein level to show cytokines effects.
Author Response
Review Report 1
Comments to the Author:
Author came up with an interesting observation of brain endothelial cell mediated neuroinflammatory pathway activation to early CSVD in patient's plasma.
I have few minor concerns,
- Sample size is very small and there is huge variability, for instance, females are outnumbered than male (PTs 9-1, CTs, 5-5, BD, %-2). Studies have shown that neuroinflammatory pathways varied based on sex. At this point we cannot make any statements based on small sample size. Reply: We thank the reviewer for the suggestion. We pointed out in the discussion the possibility of a sex-related effect. We also added few comments at the end of the discussion section, to clarify these issues as “limitations” of the study.
- Figure 2 is very hard to see, please try to improve the quality of image and make sure its legends are readable. Reply: We thank the reviewer for the suggestion. We have modified Figure 2 by improving the quality of the image.
- Based on RNA-seq, author explored qPCR of CXCL8, LIF and IL-12A, but why the author did not evaluate concentration using ELISA? An increased mRNA expression does not confirm an increased protein level to show cytokines effects. Reply: We thank the reviewer. Our hypothesis is that in CSVD patients the cerebral endothelium is involved in the development of the disease. To investigate this, we evaluated the global effect of patients' plasma on brain endothelial cell gene expression with RNA-seq. Normally, RNA-seq results need to be confirmed by a different approach (Mio, C., Conzatti, K., Baldan, F., Allegri, L., Sponziello, M., Rosignolo, F., Russo, D., Filetti, S., Damante, G."BET bromodomain inhibitor JQ1 modulates microRNA expression in thyroid cancer cells". Oncology Reports 39, no. 2 (2018): 582-588. https://doi.org/10.3892/or.2017.6152). qPCR is a widely used technique for gene expression assessment, therefore we chose to analyze the expression of three significative molecules (CXCL8, LIF and IL-12A) to confirm the sequencing data. These preliminary results describe only the pathways activated when brain endothelial cells were stimulated with plasma patients. We agree with the reviewer: an increase in mRNA expression does not confirm an increase in protein level to show cytokine effects, but that was not our focus.
Reviewer 2 Report
Comments and Suggestions for Authors
The submitted article describes scientifically sound experimental study. The presented results are of interest to researchers in the field of experimental neurology and stroke research. However, it can be improved.
1. The submitted article presents the results of a study aimed to establish if plasma taken from the asymptomatic CSVD patients contains specific markers of the disease. To do that the authors chose to reveal markers affecting brain endothelial cells, i.e. markers presumably related to CSVD pathogenesis. The authors started with comparing the transcriptomes of brain endothelial cells cultured in the presence of plasma taken from asymptomatic CSVD patients and two groups of control humans and revealed a bunch of genes differentially expressed in the three groups. In some cases they confirmed transcriptomic data with PCR data. On this basis they speculate about the possible involvement of different regulatory cascades in the effects of patient plasma and make assumptions about the markers present in the plasma. However, they did not check the latter assumptions experimentally, by direct measurements of their concentrations in plasma samples. This is a big flaw making the whole presented study much less valuable. I would suggest to find more data in the literature and ad them to the discussion.
2. The authors failed to clearly explain the goal of the study and the experimental setup at the end of Introduction .
Author Response
The submitted article describes scientifically sound experimental study. The presented results are of interest to researchers in the field of experimental neurology and stroke research. However, it can be improved.
- The submitted article presents the results of a study aimed to establish if plasma taken from the asymptomatic CSVD patients contains specific markers of the disease. To do that the authors chose to reveal markers affecting brain endothelial cells, i.e. markers presumably related to CSVD pathogenesis. The authors started with comparing the transcriptomes of brain endothelial cells cultured in the presence of plasma taken from asymptomatic CSVD patients and two groups of control humans and revealed a bunch of genes differentially expressed in the three groups. In some cases, they confirmed transcriptomic data with PCR data. On this basis they speculate about the possible involvement of different regulatory cascades in the effects of patient plasma and make assumptions about the markers present in the plasma. However, they did not check the latter assumptions experimentally, by direct measurements of their concentrations in plasma samples. This is a big flaw making the whole presented study much less valuable. I would suggest to find more data in the literature and add them to the discussion. Reply: We thank the reviewer for the suggestion. In our previous paper, we measured several molecules in the plasma of CSVD patients (doi:10.1038/s41598-019-50778-w.). Given the results, we then thought to evaluate the global effect of plasma from asymptomatic patients on brain endothelial cells. We then sought to test our hypothesis, which is that in plasma asymptomatic patients there may be a combination of molecules that stimulate cerebral endothelial cells. We implemented the introduction by clarifying our point of view.
- The authors failed to clearly explain the goal of the study and the experimental setup at the end of Introduction. Reply: We thank the reviewer. We improved the introduction.
Reviewer 3 Report
Comments and Suggestions for Authors
The patient sample is quite gender-imbalanced. Could the observed effect be female-specific?
qPCR controls. The standard procedure implies the usage of at least two housekeeping genes, but only one was used in this study.
The primer sequences should be included in the methods section.
It would be easier to understand data, if the statistcal test would be indicated in the figure captions.
24 h is not enough to asses proliferation changes. In addition, it is not clear, how exactly the viability % was calculated.
Figure 3: The Y axis starts from 0, but the data are 'fold change over untreated cells'. I suggest ether shifting the X axis to the level 1, or adding a horizontal line at that level, as in this case, the real situation of "no effect" is the value of 1.
Figure 3. For qPCR data analysis, SD usually is replaced by standard error
Figure 3. The obtained data basicly shows that even the treatment with the healthy blood donors' plasma induces the cytokines mRNA at least 3 fold, and the difference between plasma variants is smaller compared to it. Could it be due to some mechanical stress or other similar conditions?
The exact wording used looks a bit strange in some cases. For example, the word "solely" is very rare for the scientific literature, a more typical one is the word "only". In some places, commas are missing.
Author Response
1.The patient sample is quite gender-imbalanced. Could the observed effect be female-specific? Reply: We are aware of this issue, and we thank the reviewer for raising it: we pointed out in the discussion, above all under the study’s limitations, the possibility of a sex-related effect.
2. qPCR controls. The standard procedure implies the usage of at least two housekeeping genes, but only one was used in this study. Reply: We thank the reviewer. To date, the usage of two housekeeping genes is not mandatory, then we used the 2−∆∆Ct method with GAPDH as reference. (doi: 10.1155/2017/4814987; doi: 10.1186/s13317-019-0113-9)
3. The primer sequences should be included in the methods section. Reply: We thank the reviewer. We add sequences in the methods section.
3. It would be easier to understand data, if the statistical test would be indicated in the figure captions. Reply: We thank the reviewer. We have modified as requested.
4. 24 h is not enough to assess proliferation changes. In addition, it is not clear, how exactly the viability % was calculated. Reply: We thank the reviewer. We assessed viability to exclude that 24-hour plasma treatment did not induce cell death. We used the trypan blue exclusion method to calculate viability. We add reference in methods section.
5. Figure 3: The Y axis starts from 0, but the data are 'fold change over untreated cells'. I suggest ether shifting the X axis to the level 1, or adding a horizontal line at that level, as in this case, the real situation of "no effect" is the value of 1. For qPCR data analysis, SD usually is replaced by standard error
The obtained data basicly shows that even the treatment with the healthy blood donors' plasma induces the cytokines mRNA at least 3 fold, and the difference between plasma variants is smaller compared to it. Could it be due to some mechanical stress or other similar conditions? Reply: We thank the reviewer. We used the standard deviation (SD) to emphasize the effect of the plasma of patients compared with that of BD and CTRLs (in which SD is higher). Considering the standard deviation, it is likely that the effect obtained in BD or CTRL plasma treatments is random and maybe due to some mechanical stress as the reviewer suggests.
Comments on the Quality of English Language
The exact wording used looks a bit strange in some cases. For example, the word "solely" is very rare for the scientific literature, a more typical one is the word "only". In some places, commas are missing. Reply: We thank the reviewer for the suggestion. We carefully checked all the paper and made some revisions.